# Transgenerational Cold Acclimation and Contribution of Gut Bacteria in *Spodoptera frugiperda*

**DOI:** 10.3390/insects16101052

**Published:** 2025-10-16

**Authors:** Yu Song, Guo-Yun Yu, Wei Gao, Yu-Tong Mai, Jin Xu, Wen Fu, Zhi-Xiao Zhang

**Affiliations:** 1Yunnan Provincial Key Laboratory for Conservation and Utilization of In-Forest Resource, Southwest Forestry University, Kunming 650224, China; sy3167632311@163.com (Y.S.); yuguoyun1063431541@126.com (G.-Y.Y.); 15777996006@139.com (Y.-T.M.); 2Wenshan Plant Protection and Quarantine Station, Wenshan 663099, China; gw_haim@163.com; 3Plant Conservation and Quarantine Station of Yunnan Province, Kunming 650034, China; fuwenyn@yeah.net; 4Yunnan Academy of Forestry and Grassland, Kunming 650201, China

**Keywords:** *Spodoptera frugiperda*, cold stress, cold tolerance, gut bacteria, antibiotics

## Abstract

**Simple Summary:**

Studying cold stress and adaptation provides theoretical insights for predicting and controlling pests. Temperature influences gut microbiota, which may in turn affect insect cold tolerance, though the specific mechanisms and bacteria involved remain unclear. Using multigenerational cold acclimation and 16S rDNA sequencing in *Spodoptera frugiperda*, we observed decreased larval mortality and increased pupation rate over generations, indicating cold adaptation. Acclimated adults also survived extreme cold better than controls but with reduced reproductive fitness, suggesting a survival–reproduction trade-off. Antibiotics impaired both fitness and cold tolerance in all lines and disrupted gut microbial balance. Notably, nine genera and eight species were more abundant in acclimated larvae but scarce in controls. These bacteria are potentially crucial for cold adaptation. Our findings elucidate microbiota’s role in insect environmental adaptation and support developing eco-friendly pest strategies.

**Abstract:**

The study of cold stress and adaptability can provide a theoretical basis for predicting and controlling pests. Temperature shapes gut microbiota composition, and gut microbiota may affect insect temperature tolerance. However, the underlying mechanisms and bacteria species involved in insect temperature tolerance through gut microbiota are still poorly known. In this study, using a multigenerational cold-acclimation design and 16S rDNA sequencing, we investigated the possible pattern of cold acclimation and the contribution of gut bacteria in *Spodoptera frugiperda*. Results show that during cold acclimation, larval mortality decreased and pupation rate increased with the increase of treating generations, exhibiting some sort of cold adaptation. Cold tolerance tests showed that cold-acclimated adults exhibited significantly higher survival rates under extreme cold stress than those of controls. Cold acclimation also leads to the cost of reproductive fitness, indicating some trade-offs between survival and reproduction. Antibiotic treatment significantly decreased fitness and cold tolerance not only in the un-acclimated lines but also in cold-acclimated lines. Biodiversity studies through 16S RNA sequencing suggested that antibiotic ingestion disrupted the homeostasis of gut microbes. Differential analysis at the genus and species levels further showed that there were nine genera and eight species that had remarkably higher abundance in cold-acclimated lines compared with controls. References-based functional annotation revealed that most of these bacteria may play essential roles in the cold adaptation of hosts. These results provide valuable insights into the contribution of microbiota to insect fitness and will be valuable for guiding the future development of sustainable pest management approaches.

## 1. Introduction

Temperature plays an essential role in the survival, development, and distribution of animals [1,2,3]. As ectothermic organisms, coping with temperature variations is critically important for insects [4]. Optimal temperatures accelerate metabolism, growth, and reproduction, while extreme temperatures can be lethal [1,2,3]. Cold stress can negatively affect insects’ metabolism, growth, behavior, flight efficiency, and reproduction [5]. Extreme cold causes freezing injuries, damaging cells and tissues, often resulting in death [5,6]. Understanding these thermal effects is vital for biodiversity conservation and predicting and controlling insect pests [7,8].

Previous studies commonly found that prior cold or heat exposure could significantly increase insects’ heat or cold tolerance and cause physiological and molecular changes [5,9,10], which is a strategy for insects to adapt to temperature stress [5,9,10,11]. Notably, a number of studies in different insect species have demonstrated that their tolerance to extreme temperatures can be significantly enhanced after a few generations of cold [12,13,14] or heat [10,15,16] selection, signifying a multigenerational temperature acclimation. Moreover, a large number of studies have revealed that insects adapt to cold or heat stresses through physiological and molecular mechanisms, including the upregulation of cryoprotectant (e.g., glycerol) and antifreeze production proteins, cold shock protein (Csp) and heat shock protein (Hsp) expressions, as well as membrane remodeling, metabolic regulation, and epigenetic modifications [5,6,9,17].

Gut microbiota in insects play vital roles in nutrient metabolism, immune modulation, and detoxification, and can influence insect development, reproduction, and environmental adaptation [18,19,20]. Gut microbes can contribute to insects’ nutrition through diverse ways, such as by producing nutritional components and providing digestive enzymes [19]. Studies on cockroaches and termites have demonstrated the vital contribution of gut microbiota to nutrition for hosts living on a suboptimal diet [21]. In addition to aiding in nutrient achievement, gut microbiota also benefit herbivorous insects by neutralizing plant toxic defense substances [22]. Meanwhile, these resident gut microbiota can also benefit insects by preventing the colonization of pathogens in the intestine [23]. Moreover, previous studies also found that microbiota could have positive or negative effects on the reproductive fitness of insects [24]. Thus, in fruit flies, some *Enterococcus* species (e.g., *E. faecalis*), had detrimental impacts on reproduction [25,26], while some other species, such as *Enterobacter cloacae*, *Citrobacter braakii*, *Pantoea dispersa*, and *Klebsiella pneumonia*, showed beneficial effects on reproductive fitness [25,27]. Further studies also indicated that most of these bacteria-mediated physiological changes were caused by the presence of microbiota per se and the production of metabolites by it [28]. Moreover, it is recognized that bacteria in the gut microbiota interact deeply with the host by regulating various physiological characteristics [28,29,30]. The interaction induced molecular signals can upregulate and downregulate multiple cellular pathways of the host and alter its overall physiological state [28]. Given the range of beneficial functions provided by microbiota, it may also shape the ability of their hosts to tolerate stressful situations [28,29].

Previous studies have shown that temperature significantly impacts insect gut microbiota by altering microbial composition and function [30,31]. In some insects, seasonal shifts in temperature induce behavioral and physiological changes, such as hibernation, and these changes have, in some instances, been correlated with modulations of the gut microbiota [30]. For example, the abundance of *Wolbachia* spp. increased, while that of *Pseudomonas* spp. decreased during overwintering in the spring field cricket, *Gryllus veletis* [32]. A study on bees also revealed a correlation between the composition of the gut microbiota and the hibernation behavior of bees [33]. In *Drosophila melanogaster*, there is a significant correlation between the increase in temperature and the relative abundance of Proteobacteria [34].

On the other hand, gut microbiota may influence insect thermal tolerance by modulating host physiology [30]. In humans, the presence of intestinal bacteria such as *Lactobacillus* spp. is associated with the production of Hsps, which are capable of repairing cell damage caused by heat stress [35]. Some studies have also investigated the impact of gut microbiota on insect heat tolerance [30,31]. For instance, some heritable symbionts, such as *Hamiltonella* spp. and *Serratia* spp., can enhance the heat tolerance of their aphid hosts [36]. In *D. melanogaster*, disruption of gut microbiota did not affect the heat tolerance of flies but significantly decreased their cold tolerance [29]. In *Bactrocera dorsalis*, gut microbiota increased the adult’s cold tolerance, possibly by stimulating the metabolic pathways of proline and arginine in these flies [37]. A recent study in the termite *Reticulitermes flavipes* showed seasonal adaptation to cold and an increased richness of *Methanobrevibacter* spp., which may indicate a shift in metabolism related to overwintering from acetogenesis to methanogenesis [38]. Therefore, gut microbiota can alter the host’s metabolism, energy reserves, and gene expression, which may indirectly affect the temperature tolerance of hosts [30,39].

The fall armyworm (FAW), *Spodoptera frugiperda* (Lepidoptera: Noctuidae), is a highly destructive invasive pest native to tropical and subtropical regions of the Americas [40]. It poses a significant threat to global food security due to its rapid spread, high reproductive capacity, and broad host range. FAW was first detected in China in January 2019 [41]. Since then, it has spread across over 20 provinces, causing severe agricultural damage. Its resistance to conventional pesticides further complicates the control efforts of this moth pest [42,43,44]. It has been shown that the thermo-tolerance of FAW adults is associated with sex, age, and mating status [45]. Prior cold acclimation enhanced the cold tolerance of larvae [46], and prior heat acclimation increased the heat tolerance of larvae and adults [47]. A number of studies have also revealed the gut microbial community in FAW populations from different hosts and different regions (e.g., [48,49]) and their modulating effect on plant defense responses [50], energy and metabolic homeostasis [51], and development and reproduction [52] of the host.

Based on the above achievements on temperature stress and adaptation mechanisms in insects, we hypothesize that temperature shapes gut microbiota composition, and gut microbiota contributes to the cold tolerance of FAW. To test this hypothesis, we investigated and discussed the possible pattern and significance of the cold acclimation of FAW and the contribution of gut bacteria by using a multigenerational cold-selection design, antibiotic treatments, and 16S rDNA sequencing.

## 2. Materials and Methods

### 2.1. Insects

FAW larvae were collected from corn plants in a corn field near Dongchuan Town, Yunnan Province of China. Subsequently, the larvae were reared under conditions of 27 ± 1 °C, relative humidity of 60–80%, and a light dark cycle of 14:10 h (set 9:00 am to 8:00 pm as scotophase, and the other time as photophase) on an artificial diet [53]. According to the characteristics of the abdomen, mature pupae are differentiated by gender [54], and male and female pupae are stored separately in cages. To ensure the virgin status and age consistency of experimental individuals, the emergence of adult insects is recorded daily, and the male and female adult insects are housed separately in cages. The insects were fed 10% honey solution during the adult stage.

### 2.2. Cold Acclimation

This study was carried out over seven consecutive generations (L1–L7) of low-temperature acclimation treatment on the complete life cycle of FAW (Figure 1). The specific operation is as follows: the first three generations (L1~L3) undergo a single low-temperature treatment (referred to as treatment LT) at 11 °C for 8 h daily; subsequent generations (L4~L7) adopt two treatment plans: daily 9 °C for 8 h (LT1) or 11 °C for 12 h (LT2). All low-temperature treatments were carried out during the dark period except LT2, where 10 h were in the dark period and 2 h were in the following light period. The standard cultivation temperature of 27 °C was maintained for the rest of the time (under the same cultivation conditions as the above). A control group (CK) was established, which was reared continuously for seven generations at a constant temperature of 27 °C. All treatment groups of insects were fed with the same feed scheme as described above.

Measurement of development and fitness of different generations. We used 60 individuals (n = 60) per sample to measure the developmental stages of larvae, male pupae, and female pupae. We measured the weight of the 6th instar larvae and 3 d-old male or female pupae using a Sartorius MSX electronic balance (Sartorius, Göttingen, Germany), accurate to 0.0001 g, with 60 insects (n = 60) per sample. We also measured the mortality of larvae and the eclosion rate of pupae by setting 5 replicates for each treatment, with 24 individuals per replicate.

The weight of 3 d-old male and female adults, including whole body weight and abdomen weight, was measured as above. Adults’ body length, abdominal length, and abdominal width (maximum width of the abdomen from one side to the other) were measured under a stereomicroscope (Olympus, Tokyo, Japan) equipped with an ocular micrometer. Thirty males or females were used for each measurement (n = 30).

Then, 3 d-old adults (sexually mature) were collected and paired in plastic cages (8 cm in height, 15 cm in width, 25 cm in length), with one pair per box, for mating and egg laying. Each cage comes with a serrated folded paper strip (15 × 20 cm) as the spawning substrate and 10% honey solution as food. Eggs laid within the first five days after mating (>90% of the total eggs were laid within 5 d after mating [53]; in order to save time, we did not measure the reproductive capacity throughout their lifetime, as some females have a oviposition period greater than 10 d) were collected and incubated in a culture dish (8.5 × 1.5 cm) under the above conditions. The number of hatched eggs after 4 days of incubation was recorded. Fifteen pairs (n = 15) were used for each sample to measure egg production and hatching rate.

### 2.3. Survival of the Cold-Acclimated Adults Exposed to Cold Stress

The 3 d-old male and female adults from each generation of the above treatments were collected and were subjected to cold stress at −5 °C for 2 h. Mortality was assessed 24 h post-treatment under a constant temperature of 27 °C. Three replicates were used for each treatment, with 8 males or females being used for each replicate.

### 2.4. Effect of Antibiotic Treatment on Fitness and Cold Acclimation

According to previous research [26,52,55], four antibiotics, including ampicillin, streptomycin, tetracycline, and metronidazole, were used to disrupt the composition of gut microbiota. The sixth generation (L6) larvae from lines CK, LT1, and LT2 were collected and fed with an artificial diet to which the four antibiotics had been independently added from the second instar for 8 d (other times, they were fed with a normal diet as above), at a concentration of 1.5 mg of each antibiotic per 1 g of diet, forming treatment CKa, LT1a, and LT2a. To exclude the negative effect of antibiotic treatment on insects, the offspring from the antibiotic-treated insects were reared on the artificial diet and 10% honey solution without antibiotics, under the same rearing conditions as above. Biological analyses were the same as above. The L6 and L7 of lines CK, LT1, and LT2 were used as controls.

### 2.5. Effect of Cold Acclimation and Antibiotic Treatment on the Composition of Gut Bacteria

The sixth instar larvae from the L6 of lines CK, CKa, LT1, LT1a, LT2, and LT2a were sampled and stored at −80 °C before use. Total genome DNA was extracted from samples using the CTAB/SDS method. DNA purity and concentration were examined by 1.0% agarose gel electrophoresis. The 16S rDNA regions were amplified via PCR using the V3-V4 primers (515F: 5′-GTGYCAGCMGCCGCGGTAA-3′; 806R: 5′-GGACTACNNGGGTATCTAAT-3′) and the Phusion High-Fidelity PCR Master Mix (New England Biolabs, Ipswich, MA, USA). Libraries were prepared using the TruSeq DNA PCR-Free Library Preparation Kit (Illumina, San Diego, CA, USA), and index codes were added following the manufacturer’s protocols. The libraries were subjected to sequencing on an Illumina NovaSeq PE250 platform (Illumina, USA), yielding 250 bp of paired-end reads.

The following data analyses were conducted using QIIME2 (Microbial Ecological Quantitative Insights) pipeline [56]. The generated reads were denoised and chimeric sequences were deleted using the DADA2 plugin [57] to obtain amplicon sequence variants (ASVs). ASVs were aligned using the Align-to-tree-mafft-fasttree pipeline [58], and a phylogeny for ASVs was constructed using q2-phylogeny [59]. Species annotation was conducted for each ASV using QIIME2 classify-sklearn [56,60] and the pre-trained Naive Bayes classifier, with the annotation database being Silva 138.1.

Alpha diversity was assessed using QIIME2. The calculated indices included Observed species, Good’s coverage, Shannon, Simpson, and Chao1. Then Bray–Curtis distance metrics were used to assess beta diversity; the significance of differences between groups was tested using NPMANOVA (non-parametric multivariate analysis of variance) and visualized using PCoA (Principal coordinates analysis) plot.

Linear discriminant analysis (LDA) of the effect size was used to determine ASVs that discriminated among the populations with an LDA score of more than 4.0.

BugBase was used to predict the possible functional annotation of taxa in different size fractions, which predicts functions of uncultured prokaryotes from the known functions of cultured bacterial genera.

### 2.6. Statistics

All data on the developmental period, body weight/size, reproduction, and survival rate are presented as mean ± SE. Percentage data were subjected to arcsine square-root transformation to achieve normality. Prior to analysis, the homogeneity of variances and normality were verified using Levene’s test and the Shapiro–Wilk test, respectively. Subsequently, the data were analyzed by one-way ANOVA, followed by Fisher’s LSD test for multiple comparisons. All statistical analyses were performed using SPSS 16.0, with a significance level of α < 0.05.

## 3. Results

### 3.1. Effect of Cold Exposure on the Fitness of FAW

Cold exposure significantly (*p* < 0.05; Figure 2a–e; Appendix A) prolonged the developmental period of the larvae and pupae and the lifespan of male and female adults for almost all the test generations (L1, L3~L7; except the lifespan of male adults from L3 and L6). It is also obvious that the developmental period of larvae and pupae increased with the decreasing temperature (or the increase of cold exposure duration) and the increase in generations (L1~L7). Cold exposure lines showed significantly (*p* < 0.05; Figure 2f–j) heavier body weight in larvae, male and female pupae, and adults than those of controls in almost all measurements (except L7 male adults). Cold exposure lines also showed significantly heavier adult abdominal weight, and larger body size of adults, including male and female body length, abdominal length, and width in most of the measurements (45/48; Appendix A).

Cold exposure also significantly (*p* < 0.05; Figure 3a,b; Appendix A) increased the mortality of larvae and reduced the eclosion rate of pupae in half of the measurements. Similarly, the mortality of larvae increased and the eclosion rate decreased with the decreasing of temperature and the increasing of generations.

Reproduction test showed cold exposure lines had sigificantly (*p* < 0.05; Figure 3c,d; Appendix A) lower numbers of eggs laid and egg-hatching rates in most of the test generations (except for the number of eggs laid in L5 and egg-hatching rate in L7).

Moreover, the cold tolerance test showed that cold-acclimated female and male adults exhibited significantly (*p* < 0.05; Figure 3e,f; Appendix A) higher survival rates under extreme cold stress (−5 °C for 2 h) at L3, L6, and L7 generations than those of controls. It was also noticed that the cold tolerance of the cold-exposed lines increased with lower exposure temperatures, longer individual exposure durations, and more treatment generations.

### 3.2. Effect of Cold Exposure and Antibiotic Ingestion on the Fitness of FAW

Antibiotic ingestion in the L6 generation significantly (*p* < 0.05; Figure 4A, Appendix A) reduced the body weight and size of the larvae, pupae, and adults both in CK and cold exposure lines in most comparisons. This treatment also had significantly negative effects on reproductive fitness in most cases (*p* < 0.05; Figure 4A). Antibiotics had no apparent effects on the developmental period of larvae and pupae, as well as the lifespan of female and male adults (*p* > 0.05; Figure 4A). Moreover, antibiotic treatment significantly reduced male and female cold tolerance both in CK and the cold exposure lines (*p* < 0.05; Figure 4A).

Similarly, the offspring (L7, no antibiotic treatment in this generation) of antibiotics-treated lines (treated in L6 generation) also showed lower fitness than those of controls (Figure 4B, Appendix A).

### 3.3. Effect of Antibiotics Feeding on the Composition of Bacteria in Different Life Stages

#### 3.3.1. Sequencing and Quality Control

The 16S rDNA sequencing achieved 74,500~179,500 clean reads from each of the 24 samples, with an average read length of 416~429 bp, and the Q20 and Q30 being 98.54~98.98% and 94.91~96.21%, respectively (Appendix A). Analyzing the dilution curves showed that all of them reached a plateau (Appendix A), suggesting an adequate sequencing depth and breadth for all samples. Good’s coverages were all 100% (Appendix A), indicating that the samples were adequate to provide a sufficient estimation of bacteria diversity in FAW.

#### 3.3.2. Diversity Indices of Bacterial ASVs

A total of 1155 different ASVs were obtained from the sequencing (Appendix A). The number of ASVs ranged from 62 to 195 among different samples (Appendix A). Among the total samples, very few (47/1155) ASVs were present in all samples (Figure 5a); relatively more common ASVs showed in the non-antibiotic-treated groups (69) and antibiotic-treated groups (66) (Figure 5b,c). Accordingly, the observed features (number of ASVs) and Shannon’s and Chao’s alpha diversity indices reflected the difference between groups (Figure 5d–f; Appendix A). Beta diversity analysis based on Bray–Curtis distances revealed that there was a significant difference between samples (PMANOVA: *F*_5,18_ = 9.867, *p* < 0.001; Figure 5g–i).

#### 3.3.3. Taxonomy Assignment and Abundance Analysis

The obtained 1155 ASVs (Appendix A) were annotated to 20 phyla (Appendix A), 36 classes (Appendix A), 80 orders (Appendix A), 133 families (Appendix A), 213 genera (Appendix A) and 71 species (Appendix A). At the phylum level (Figure 6a), Firmicutes and Proteobacteria were the first and second predominant phyla in CK; Firmicutes was also the first dominant phylum in LT1 and LT2, although its abundance decreased obviously, while the second dominant phylum was Actinobacteriota. In antibiotic groups, Proteobacteria and Firmicutes were the first and second predominant phyla in CKa; Firmicutes was the first dominant phylum of LT1a and LT2a, with Proteobacteria being the second dominant phylum of LT1a and Actinobacteriota being the second dominant phylum of LT2a.

At the family level (Figure 6b), in the non-antibiotic groups, the predominant family was Enterococcaceae for CK, whereas it was Corynebacterium for LT1 and Lactobacillaceae for LT2. In the antibiotic groups, Enterococcaceae and Morganellaceae were the first and second families of CKa, Enterococcaceae and Enterobacteriaceae for LT1a, Corynebacterium and Enterococcaceae for LT2.

At the genus level (Figure 6c), the predominant genus was *Enterococcus* for CK, but it was *Corynebacterium* for LT1 and *Weissella* for LT2. In the antibiotic groups, the predominant genus was *Proteus* for CKa, but it was *Enterococcus* for LT1 and *Weissella* for LT2.

At the species level (Figure 6d), *Weissella viridescens* was the predominant species for CK, LT1a, LT2, and LT2a, while *Corynebacterium stationis* was the predominant species for CKa and LT1.

Moreover, LDA also demonstrated obvious differences on the biomarkers between different treatments (Figure 7a).

#### 3.3.4. Functional Prediction

The functional prediction by BugBase showed remarkable differences among treatments (Figure 7b). The predominant function for CKa was Contains_Mobile_Elements (17.46%), while for other treatments all were Gram_Positive but showed large scale variation on relative abundance (24.33~61.26%). The second and third dominant functions can be Contains_Mobile_Elements, Forms_Biofilms, Aerobic, or Facultatively_Anaerobic, and showed notable differences on relative abundance between treatments (9.04~36.76%).

#### 3.3.5. Differential Analysis and References-Based Functional Annotation on the Genus and Species Levels

Further differential analysis showed that there were nine genera, including *Corynebacterium*, *Staphylococcus*, *Myroides*, *Serratia*, *Carnobacterium*, *Jeotgalicoccus*, *Providencia*, *Glutamicibacter*, and *Paenochrobactrum*, that had remarkably higher abundances in LT1 and/or LT2 but lower abundances in the control and antibiotic-treated lines (CK, CKa, LT1a, LT2a). The total abundance of these species was 0.52 for LT1 and 0.18 for LT2, while it was lower than 0.10 for controls. Differential analysis showed that there were eight species, including *Corynebacterium stationis*, *Carnobacterium maltaromaticum*, *Corynebacterium variabile*, *Myroides profundi*, *Microbacterium amylolyticum*, *Ignatzschineria indica*, *Koukoulia aurantiaca*, and *Acinetobacter gerneri,* that had remarkably higher abundances in LT1 (0.053) but lower abundances in other treatments (<0.036).

Functional annotations based on references indicate that most of these genera contained some cold/stress tolerant species, and some of them may contribute to the cold/stress tolerance and acclimation of hosts (Table 1).

## 4. Discussion

In FAW, a previous study had shown that the developmental periods of different stages were negatively correlated with temperatures within 18~32 °C [73]. In the present study, cold exposure significantly prolonged the developmental period of larvae and pupae, and the lifespan of adults, with the developmental period of larvae and pupae increasing with the decreasing of temperature and the increasing of treating generations. This may be because low temperatures slow insect metabolism, delaying development and reducing growth rates [2,3]. Moreover, cold exposure lines showed significantly heavier body weight of larvae, male and female pupae and adults, as well as larger body sizes and abdominal sizes of male and female adults, than those of controls in all generations. Studies on insects generally indicate that body size increases under lower temperatures [74]. This is attributed to an extended larval (feeding) stage in combination with a reduced metabolic rate, which together promote greater nutrient accumulation. It is generally found that the reproductive fitness of female insects is significantly correlated to their body size [75,76], including FAW [77]. However, reproduction test showed that cold exposure lines had significantly lower numbers of eggs laid in all seven generations and egg-hatching rates in the first five generations. It was also shown that low temperatures improved adult body size but reduced their fecundity in the greater wax moth, *Galleria mellonella* [78]. Cold stress-incurred reproductive costs have also been found in many other insect species, which may be due to cold causing freezing injuries, damaging cells and tissues [5,6], as well as trade-offs between winter survival and reproduction under cold conditions and limited resources [79].

Notably, during cold acclimation treatments, the mortality of larvae decreased and the pupation rate increased with the increasing of treating generations, exhibiting some kind of cold adaptation. Moreover, the cold tolerance test showed that cold-acclimated female and male adults exhibited significantly higher survival rates under extreme cold stress in L3, L6, and L7 generations than those of controls. It is widely exhibited that temperature acclimation is able to modify insects’ cold or heat tolerance due to physiological and molecular adaptation mechanisms [5,9,10,11]. For instance, in the ragweed leaf beetle, *Ophraella communa*, cold temperatures enhanced cold tolerance in the next generation [12]. In the grain aphid *Sitobion avenae*, lower temperature (from 20 °C to 10 °C) cultures for three successive generations showed a gradual increase in cold hardiness [13]. As a native of subtropical and tropical Americas [40], the global expansion of FAW is primarily constrained by temperature. This is particularly relevant for its establishment in China, which is largely situated in the northern temperate zone [80]. Despite this climatic barrier, FAW has successfully overwintered in several southern provinces, including Yunnan, Guangxi, Hainan, and Guizhou, and continues to exhibit a northward expansion trend [80]. The present study also revealed that the colder the exposure temperature and the greater the number of generations, the stronger the cold tolerance observed in the selected lines was, suggesting a fast and high potential adaptation of FAW to cold environments. Such a strong adaptive feature may allow FAW to spread more widely across China.

Previous research indicates that gut microbiota are critical in shaping host thermo-tolerance [28,30,39]. However, the underlying mechanisms and bacterial species involved in insect temperature tolerance through gut microbiota are still poorly known. In the present study, we show that multigenerational cold selection changed the diversity (Shannon) and richness (Chao) of gut bacteria in FAW larvae. Beta diversity analysis further showed a significant difference between treatments on the composition of gut bacteria. Moreover, both alpha and beta analyses showed antibiotic treatment remarkably changed the diversity and abundance of gut bacteria. Accordingly, fitness tests showed that antibiotic treatment negatively affected the development and reproduction in the treated generation (fed antibiotics during the larval stage) and their untreated offspring (without feeding antibiotics). Statistical analysis further showed that antibiotic treatment decreased the cold tolerance not only in the un-acclimated lines (CK vs. CKa) but also in cold-acclimated lines (LT1 vs. LT1a, and LT2 vs. LT2a) in both tested generations. These results have suggested that antibiotic ingestion disrupted the homeostasis of gut microbes, which then negatively affected the fitness and cold tolerance of FAW. Similarly, disruption of gut microbiota significantly decreased the cold hardiness in fruit flies [29]. In FAW, previous studies have shown that gut bacteria play roles in plant-defense responses [50], metabolic homeostasis [51], and the development and reproduction [52] of the host. These results collectively confirm the significance of gut bacteria in the fitness and stress response of FAW. Due to their crucial role in regulating insect survival and reproduction, symbiotic microbes have been proposed as a potential target for controlling agricultural pests [52,81,82]. Therefore, modifying symbiotic gut microbes presents a promising strategy for managing FAW populations.

Taxonomic and functional analyses also showed remarkable differences between treatments. Results showed Firmicutes and Proteobacteria were the predominant phyla, and Enterococcaceae, Lactobacillaceae, Corynebacterium, and Enterobacteriaceae were the predominant families, while their relative abundance and richness are significantly different between treatments. For example, Firmicutes and Proteobacteria were the first and second predominant phyla in CK, while in LT1 and LT2, the second dominant phylum was Actinobacteriota. Other studies also found that Firmicutes and Proteobacteria are predominant phyla not only in FAW [83] but also in many other insect species [84,85]. Moreover, software-based functional prediction by BugBase showed remarkable differences among treatments, and LDA also demonstrated obvious differences on biomarkers between different treatments, further confirming that cold acclimation and antibiotics had significantly altered the composition and functions of gut bacteria in FAW.

Differential analysis at the genus and species levels further showed that there are nine genera, and eight species had remarkably higher abundance in LT1 or LT2 but lower abundance in other treatments (CK, CKa, LT1a, and LT2a). Reference-based functional prediction revealed that two of these genus, namely *Carnobacterium* and *Microbacterium*, contained some psychrotrophic bacteria species, such as *C. alterfunditum*, *C. maltaromaticum* (also showed in this study), and *C. jeotgali* [66], and *M. phyllosphaerae* [64]. Kosiorek et al. [66] identified three *Carnobacterium* species that fermented a wide range of carbohydrates, alcohols, and organic acids, which may have played roles in functions such as cold tolerance and antimicrobial activities. Zhang et al. [64] characterized a novel cold-adapted GH15 GA-like trehalase (designated MpTre15A) from the psychrotolerant bacterium *M. phyllosphaerae* LW106, and showed its important function in the accumulation of trehalose and cold adaptation. The left six genera also contained some cold/stress tolerant bacteria species. For example, in the flesh fly *Sarcophaga peregrine*, host heat tolerance shapes the community structure of gut microbiota, which in turn modulates the fitness of the host; heat stress increased the abundance of *Wohlfahrtiimonas* but decreased the abundance of *Ignatzschineria* in larvae [71]. In the present study, we found a relatively higher abundance of *Ignatzschineria* spp in larvae. In *Bactrocera dorsalis*, *Providencia* is one of the predominant bacteria genera and may function in host fitness and adaptation [37,86]. *Acinetobacter calcoaceticus* is a cold-tolerant bacteria, which can produce unsaturated and medium-chain fatty acids, SOD and CAT [69].

In insects, studies have shown that microbiota can play a role in host thermo-tolerance. For example, *Hamiltonella* and *Serratia* in aphids [36] and *Methanobrevibacter* spp. in termites [38] can enhance host fitness under thermal stress. In our study, we also found that *Serratia* has higher relative abundance in cold-adaptation lines but lower abundance in controls. However, no *Hamiltonella* was found in FAW larvae, and the abundance of *Methanobrevibacter* spp. were minimal. In humans, the presence of gut bacteria such as *Lactobacillus* sp. is correlated with the production of heat shock proteins, a family of chaperone molecules able to repair cellular damages generated by thermal stress [35]. In FAW, a previous study has shown that *Lactobacillus* spp. have high abundance in the reproductive organs of adults and may function in reproduction protection [83]. In the present study, however, the abundance of *Lactobacillus* spp. is minimal in all larvae from different treatments.

In addition, we observed an interesting pattern: the control population appears to be more strongly affected by antibiotics (CKa vs. CK) than cold-acclimated lines (either LT1a vs. LT1, or LT2a vs. LT2), i.e., the difference between CKa and CK is more remarkable than that between LT1a and LT1 or between LT2a and LT2. Exposure to adverse conditions, including cold stress, has been demonstrated to elevate bacterial resistance to antibiotics through mechanisms like biofilm formation and upregulation of stress–response pathways [87,88]. In the present study, BugBase-based functional prediction also showed the highest Forms_Biofilms function in LT1 cold-acclimated FAW populations. Therefore, it is possible that cold-acclimated FAW populations are more resilient to antibiotic treatment, or that their microbial communities have already adapted to stress and are therefore less susceptible to disruption. However, further research is necessary to investigate this hypothesis and the underlying mechanisms. In summary, this study demonstrated that multigenerational cold acclimation significantly enhanced cold tolerance in *S. frugiperda*, which is accompanied by the enrichment of specific gut microbiomes that may play roles in the cold adaptation of hosts. Disruption of gut microbiota impaired the cold tolerance in both cold-acclimated lines and un-acclimated lines, confirming the essential role of gut microbiota. These findings highlight the microbiota’s contribution to insect environmental adaptation and offer insights for sustainable pest control strategies.

## Figures and Tables

**Figure 1 insects-16-01052-f001:**
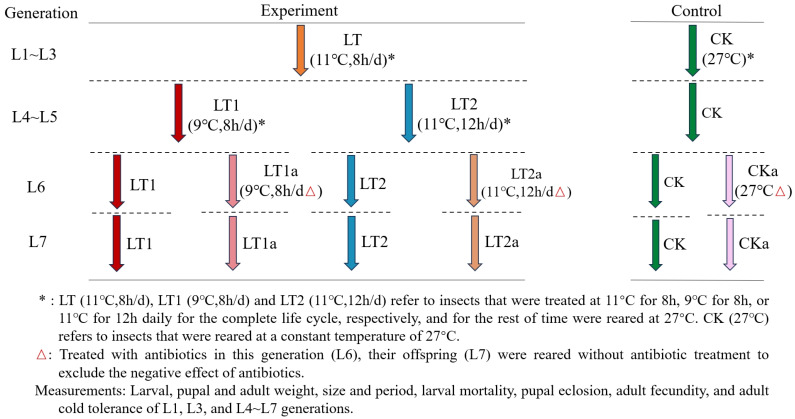
Experimental design. *Spodoptera frugiperda* was acclimated under cold conditions for seven generations, and its development and fitness were measured in different generations.

**Figure 2 insects-16-01052-f002:**
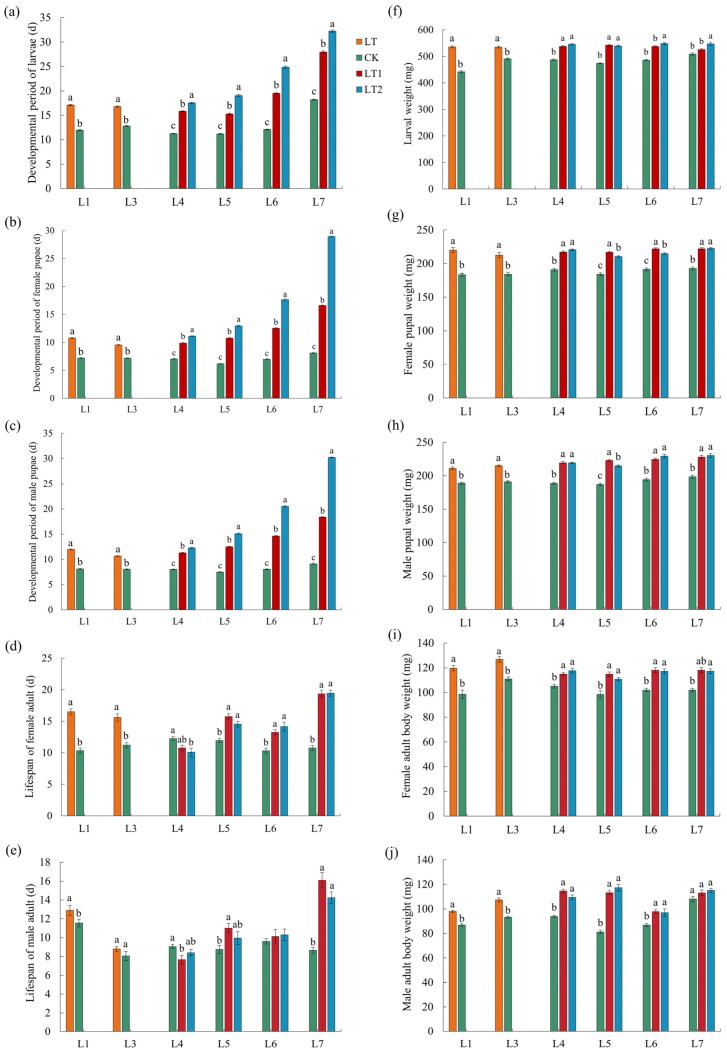
Effect of cold exposure on the development and body weight of FAW. (**a**) developmental period of larvae; (**b**) developmental period of female pupae; (**c**) developmental period of male pupae; (**d**) lifespan of female adults; (**e**) lifespan of male adults; (**f**) body weight of larvae; (**g**) body weight of female pupae; (**h**) body weight of male pupae; (**i**) body weight of female adults; and (**j**) body weight of male adults. In each generation of each subgroup, bars with different letters are significantly different (*p* < 0.05; analyzed by ANOVA followed by LSD test for multiple comparisons). The same color scheme as Figure 1 is used for easy comparison.

**Figure 3 insects-16-01052-f003:**
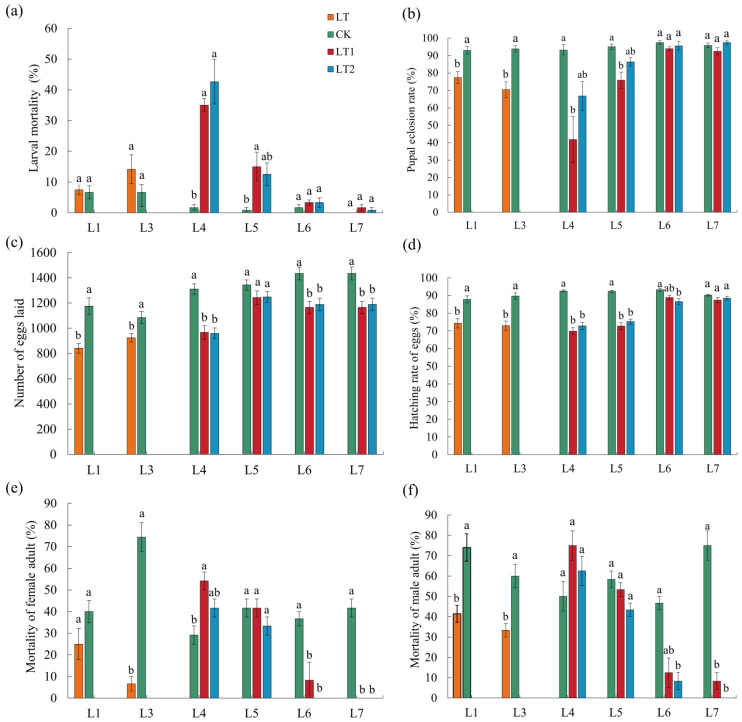
Effect of cold exposure on the survival and reproductive fitness of FAW. (**a**) Larval mortality; (**b**) pupal eclosion rate; (**c**) number of eggs laid; (**d**) egg hatching rate; (**e**) mortality of female adults under extreme cold stress (−5 °C for 2 h); and (**f**) mortality of male adults under extreme cold stress (−5 °C for 2 h). In each generation of each subgroup, bars with different letters are significantly different (*p* < 0.05; analyzed by ANOVA followed by LSD test for multiple comparisons). The same color scheme as Figure 1 is used for easy comparison.

**Figure 4 insects-16-01052-f004:**
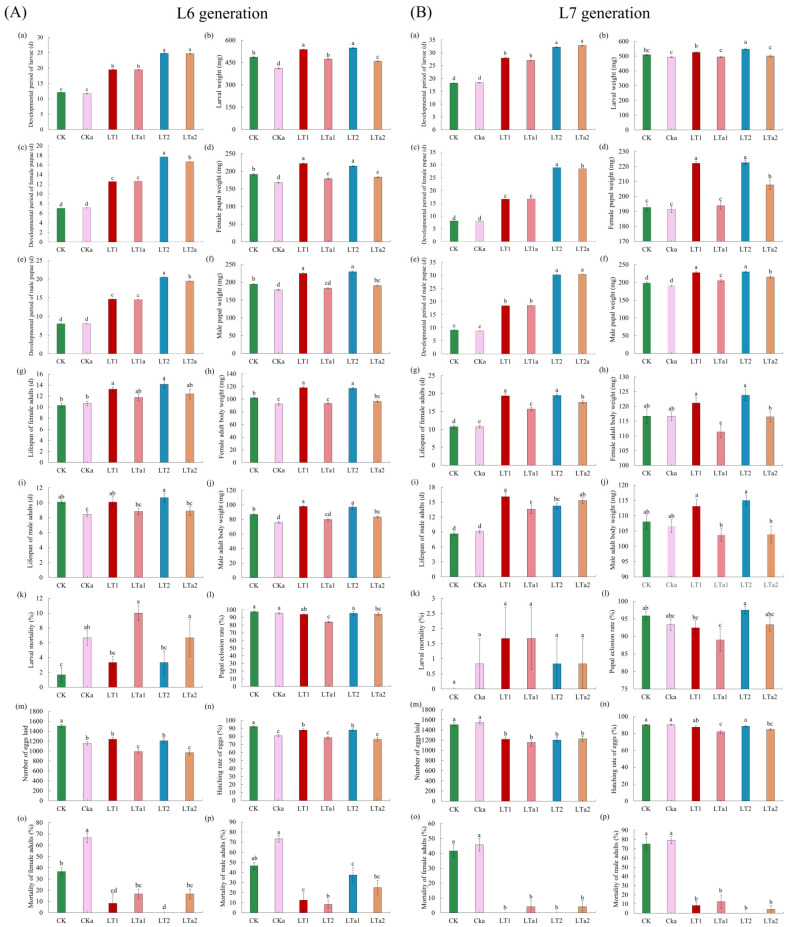
Effect of cold and antibiotic treatments on the fitness of L6 and L7 FAW. (**A**) L6 generation: (**a**) developmental period of larvae; (**b**) larval weight; (**c**) developmental period of female pupae; (**d**) female pupal weight; (**e**) developmental period of male pupae; (**f**) male pupal weight; (**g**) lifespan of female adults; (**h**) female adult body weight; (**i**) lifespan of male adults; (**j**) male adult body weight; (**k**) larval mortality; (**l**) pupal eclosion rate; (**m**) number of eggs laid; (**n**) hatching rate of eggs; (**o**) mortality of female adults under extreme cold stress (−5 °C for 2 h); and (**p**) mortality of male adults under extreme cold stress (−5 °C for 2 h). (**B**) L7 generation: the order and name of subgraphs are the same as L6 generation. In each generation of each subgroup, bars with different letters are significantly different (*p* < 0.05; analyzed by ANOVA followed by LSD test for multiple comparisons). The same color scheme as Figure 1 is used for easy comparison.

**Figure 5 insects-16-01052-f005:**
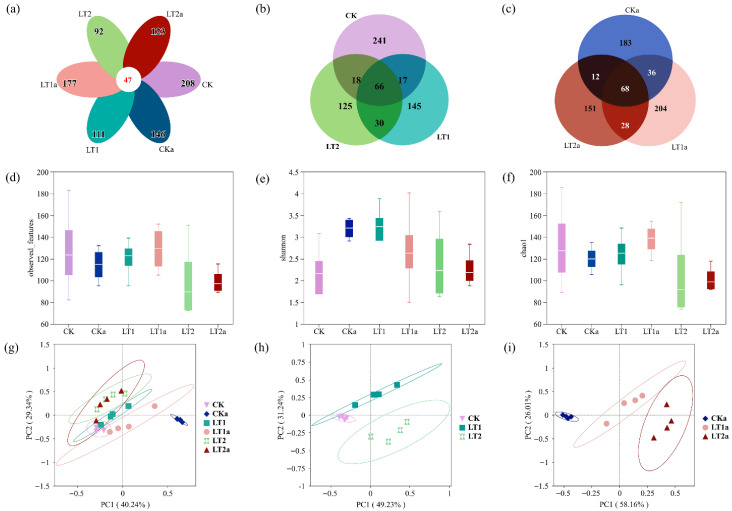
The ASVs Venn diagrams and alpha and beta diversity analyses. (**a**) ASVs Venn diagram of all samples; (**b**) ASVs Venn diagram of non-antibiotic-treated groups; (**c**) ASVs Venn diagram of antibiotic-treated groups; (**d**) observed features; (**e**) Shannon’s index; (**f**) Chao’s index; (**g**) PCoA ordination based on Bray–Curtis distances of all samples; (**h**) PCoA ordination based on Bray–Curtis distances of non-antibiotic-treated groups; and (**i**) PCoA ordination based on Bray–Curtis distances of antibiotic-treated groups.

**Figure 6 insects-16-01052-f006:**
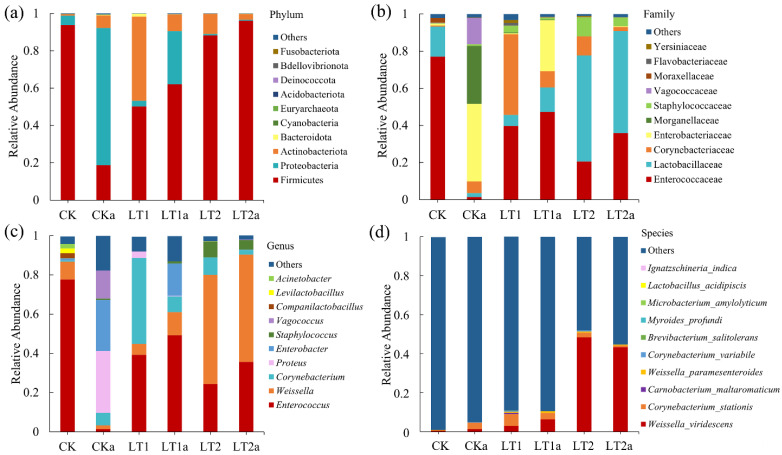
Taxonomic assignment of bacterial ASVs at the phylum (**a**), family (**b**), genus (**c**), and species (**d**) levels in different samples (top 10).

**Figure 7 insects-16-01052-f007:**
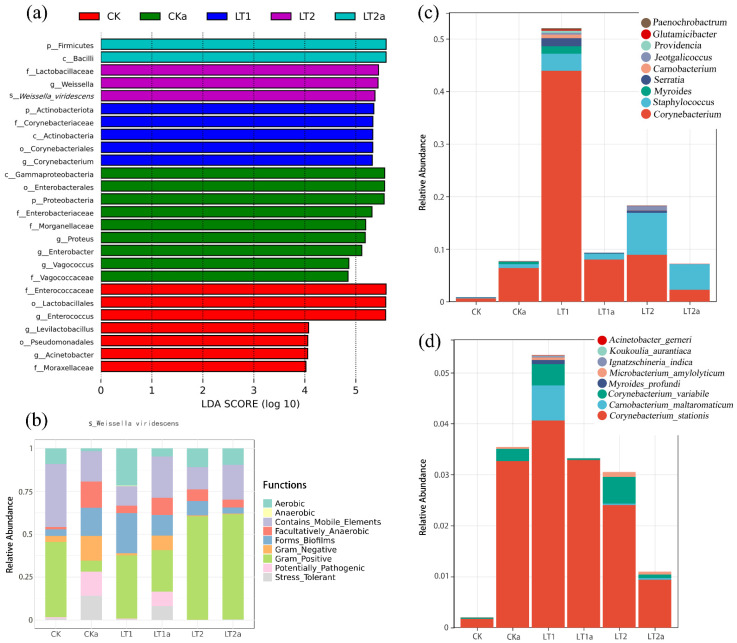
Biomarkers revealed by LDA (**a**), functional prediction by BugBase (**b**), and differential analysis of abundance at the genus (**c**) and species (**d**) levels in different samples.

**Table 1 insects-16-01052-t001:** Bacteria that may contribute to the cold acclimation of the host.

Taxonomy				Functional Annotation	Reference
Phylum	Family	Genus	Species		
Actinobacteriota	Corynebacteriaceae	*Corynebacterium*	*C. stationis*, *C. variabile*	Thermal and solvent stress tolerance	[61,62]
Actinobacteriota	Micrococcaceae	*Glutamicibacter*		Cold-tolerant bacteria, cold adaptation	[63]
Actinobacteriota	Microbacteriaceae	*Microbacterium*	*M. amylolyticum*	Psychrotolerant; produce trehalose	[64]
Bacteroidota	Flavobacteriaceae	*Myroides*	*M. profundi*	Cold-tolerant bacteria	[65]
Firmicutes	Carnobacteriaceae	*Carnobacterium*	*C. maltaromaticum*	Psychrotrophic bacteria; antimicrobial properties	[66]
Firmicutes	Staphylococcaceae	*Jeotgalicoccus*		Stress-tolerant bacteria	[67]
Firmicutes	Staphylococcaceae	*Staphylococcus*		Cold-tolerant bacteria,	[68]
Proteobacteria	Morganellaceae	*Providencia*		Promoting host fitness under stressful conditions	[37]
Proteobacteria	Moraxellaceae	*Acinetobacter*	*A. gerneri*	Cold-tolerant; produce unsaturated and medium-chain fatty acids, SOD and CAT	[69]
Proteobacteria	Rhizobiaceae	*Paenochrobactrum*		Stress-tolerant bacteria; detoxification	[70]
Proteobacteria	Wohlfahrtiimonadaceae	*Ignatzschineria*	*I. indica*	Host temperature tolerance	[71]
Proteobacteria	Yersiniaceae	*Serratia*		Stress-tolerant bacteria	[72]

## Data Availability

The original contributions presented in this study are included in the article/Appendix A. Further inquiries can be directed to the corresponding authors.

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
