# Peer review of "Transgenerational Cold Acclimation and Contribution of Gut Bacteria in Spodoptera frugiperda"

_insects, 2025, doi:10.3390/insects16101052_

Round 1

Reviewer 1 Report

Comments and Suggestions for Authors

The manuscript by Song and coworkers entitled “Transgenerational cold acclimation and contribution of gut bacteria in Spodoptera frugiperda” describes the results of a bunch of phenotypical measurements associated to development, fitness and reproduction on several populations of the fall armyworm that have been maintained in the laboratory in control conditions and under consecutive generations of low-temperature acclimation and that, additionally, have been challenged with antibiotics. The authors aim to test the hypothesis that the temperature shapes de gut microbiota composition in this insect species and that such changes contribute to the cold acclimation of the host, and they suggest that the acquired knowledge will be useful to design eco-friendly strategies to fight against this insect pest.

I find the topic very interesting, and the introduction is very convincing about the intrinsic interest of the study, but the text is hard to follow, in part due to improper English usage, in part because the way the information is presented.

Regarding the results and discussion section, the text is too descriptive. The implication of the microbiota in the cold adaptation of the insects needs to be discussed in more deep. The use of the BugBase to predict potential functions of the detected OTUs using the default phenotypes is not informative. It will be better to introduce more informative categories, as it is possible using this software. The importance of bacteria that are low in abundance in the control group, increase their presence in acclimatized individuals, and decrease due to antibiotic treatment concomitant with the loss of cold resistance, should be explored in more detail. This is where the interest of this study lies.

Furthermore, there are no final conclusions and there are no indications of how their findings can be used for pest management.

Therefore, I consider that the manuscript cannot be published in its current format and must be thoroughly reviewed before being acceptable for publication.

Below are some specific comments and suggestions that could improve some aspects of the manuscript.

Specific comments

  1. Introduction

Line 127: The hypothesis should indicate that it refers to FAW, because the hypothesis that temperature shapes gut microbiota composition has already been successfully tested in other insects.

  1. Materials and Methods

It will be useful to present a figure to schematize the experimental design, including all different generations and treatments, to be able to follow more easily the different samples analyzed.

Line 148: The authors indicate that all low-temperature treatments are carried out during the dark period, but they indicate in line 136 that the light dark cycle is 14:10 h, and the LT2 treatment is of 11ºC for 12 h. Please correct.

Line 214: OTUs or ASVs?

  1. Results

Lines 255-256: The repeated use of “increase” and “decrease” in the same sentence makes it confusing. This is just an example, as it happens several times in the text. I recommend to look for synonyms.

3.2 subsection. There is no need to start every sentence with “antibiotics ingestion”. It would be advisable to look for a more attractive writing style.

Table 1 is very difficult to read. It could de presented as a supplementary table, and the results will be more easy to interpret if they are presented in a Figure, similar to figure 2.

Figure 3: Use the same colors for each sample in all panels to facilitate interpretation.

Figure 5: I do not understand the relative abundance values in panels (c) and (d), and they do not appear mentioned in the main text.

Line 342: Should they be “LT1 and/or LT2”?

Comments on the Quality of English Language

English language usage must be corrected to reach a standard for publication. There are many typographical errors (e.g. temparature, biomarks, significatnly, adualts, incured…), use of odd expressions, misuse of verb forms and tenses.

Author Response

Comments and Suggestions for Authors

The manuscript by Song and coworkers entitled “Transgenerational cold acclimation and contribution of gut bacteria in Spodoptera frugiperda” describes the results of a bunch of phenotypical measurements associated to development, fitness and reproduction on several populations of the fall armyworm that have been maintained in the laboratory in control conditions and under consecutive generations of low-temperature acclimation and that, additionally, have been challenged with antibiotics. The authors aim to test the hypothesis that the temperature shapes de gut microbiota composition in this insect species and that such changes contribute to the cold acclimation of the host, and they suggest that the acquired knowledge will be useful to design eco-friendly strategies to fight against this insect pest.

I find the topic very interesting, and the introduction is very convincing about the intrinsic interest of the study, but the text is hard to follow, in part due to improper English usage, in part because the way the information is presented.

Regarding the results and discussion section, the text is too descriptive. The implication of the microbiota in the cold adaptation of the insects needs to be discussed in more deep. The use of the BugBase to predict potential functions of the detected OTUs using the default phenotypes is not informative. It will be better to introduce more informative categories, as it is possible using this software. The importance of bacteria that are low in abundance in the control group, increase their presence in acclimatized individuals, and decrease due to antibiotic treatment concomitant with the loss of cold resistance, should be explored in more detail. This is where the interest of this study lies.

Furthermore, there are no final conclusions and there are no indications of how their findings can be used for pest management.

Therefore, I consider that the manuscript cannot be published in its current format and must be thoroughly reviewed before being acceptable for publication.

Below are some specific comments and suggestions that could improve some aspects of the manuscript.

Our answer: We thank you very much for the constructive comments and suggestions to our MS. We have now revised the paper very carefully accordingly to all these comments and suggestions from all the three reviewers. Specifically, we have now provided more in-depth discussion on the BugBase results and the abundance and function of specific bacteria between controls and antibiotics treated lines (the last second paragraph of Discussion). Moreover, a summary paragraph was provided now at the end of the Discussion.

Specific comments

  1. Introduction

Line 127: The hypothesis should indicate that it refers to FAW, because the hypothesis that temperature shapes gut microbiota composition has already been successfully tested in other insects.

Our answer: Thanks and revised.

  1. Materials and Methods

It will be useful to present a figure to schematize the experimental design, including all different generations and treatments, to be able to follow more easily the different samples analyzed.

Our answer: Thanks and a figure was provided for the experimental design (now see Fig. 1)

Line 148: The authors indicate that all low-temperature treatments are carried out during the dark period, but they indicate in line 136 that the light dark cycle is 14:10 h, and the LT2 treatment is of 11ºC for 12 h. Please correct.

 Our answer: Thanks and corrected.

Line 214: OTUs or ASVs?

 Our answer: Thanks and corrected.

  1. Results

Lines 255-256: The repeated use of “increase” and “decrease” in the same sentence makes it confusing. This is just an example, as it happens several times in the text. I recommend to look for synonyms.

Our answer: Thanks and revised here and in the second paragraph of Discussion.

3.2 subsection. There is no need to start every sentence with “antibiotics ingestion”. It would be advisable to look for a more attractive writing style.

Our answer: Thanks and revised.

Table 1 is very difficult to read. It could de presented as a supplementary table, and the results will be more easy to interpret if they are presented in a Figure, similar to figure 2.

Our answer: We agree and have now presented Table 1 by a figure (now see Fig. 4). We also kept the data of Table 1 in Table S1 (b) for reference.

Figure 3: Use the same colors for each sample in all panels to facilitate interpretation.

Our answer: Thanks and revised accordingly.

Figure 5: I do not understand the relative abundance values in panels (c) and (d), and they do not appear mentioned in the main text.

Our answer: Thanks and have now cited them in 3.3.5. (now cited as Fig. 6c, Fig. 6d)

Line 342: Should they be “LT1 and/or LT2”?

Our answer: Yes and corrected.

Comments on the Quality of English Language

English language usage must be corrected to reach a standard for publication. There are many typographical errors (e.g. temparature, biomarks, significatnly, adualts, incured …), use of odd expressions, misuse of verb forms and tenses.

Our answer: We agree and have now very carefully checked and corrected the English of this paper.

Reviewer 2 Report

Comments and Suggestions for Authors

This manuscript by Song et al. investigates the influence of temperature on microbiota composition and how it contributes to cold tolerance in Spodoptera frugiperda. The authors used a sequencing approach to evaluate the impacts on microbial communities after cold treatment. The results are interesting and provide evidence of disruption of the microbial community after both cold exposure and antibiotic treatment. However, some minor changes to the way data are presented in Table 1 and the Discussion section are necessary to better highlight the main results.

Minor comments

Table 1. The data presented in Table 1 are very important and contribute to the main discussion. In fact, the authors state that “fitness tests showed antibiotics treatment negatively affected the development and reproduction in the treated generation (feeding antibiotics during the larval stage) and their untreated offspring (without feeding antibiotics).” However, the way the data are presented in Table 1 makes it very difficult to understand the changes under the antibiotic condition. My recommendation is to highlight the parameters in which the authors observed differences between the antibiotic vs. non-antibiotic groups, particularly in comparison with the controls.

Discussion section. It is important to emphasize in the Discussion section that the antibiotic treatment was used to disrupt the gut microbiota composition. This is only briefly mentioned in the Methods section, yet the effects of the treatment are clearly shown in Figures 3 and 4. Interestingly, the control population treatment (CKa) in Figure 4 seems to be more affected than LT1a and LT2a. This result is very interesting, but is not discussed. Could it be possible that cold-exposed populations are more resilient to antibiotic treatment, or that their microbial communities have already adapted and are therefore less prone to disruption?

Figure 1a (Developmental period of larvae graph). The bars showing information for L7 appear to be significantly different, but the green bar (Control group, CK) is denoted with the same letter “a” as the blue bar. Are both truly similar? It would be helpful to clarify this and to add in the figure legends which statistical tests were applied for L1–L3 and which were applied for L4–L7.

Line 60–61. In the text “heat shock Hsp expression”: Do the authors mean heat shock proteins (HSPs)? It is important to define the abbreviation here and to use it consistently throughout the text. Is the abbreviation in line 97 referring to the same proteins?

Line 256. Misspelling of “Temperature.”

Line 355. Misspelling of “Present study.”

Author Response

Comments and Suggestions for Authors

This manuscript by Song et al. investigates the influence of temperature on microbiota composition and how it contributes to cold tolerance in Spodoptera frugiperda. The authors used a sequencing approach to evaluate the impacts on microbial communities after cold treatment. The results are interesting and provide evidence of disruption of the microbial community after both cold exposure and antibiotic treatment. However, some minor changes to the way data are presented in Table 1 and the Discussion section are necessary to better highlight the main results.

Our answer: We appreciate for the positive comments and constructive suggestions to our MS. We have now revised our MS very carefully accordingly to all these comments and suggestions from all the three reviewers.

Minor comments

Table 1. The data presented in Table 1 are very important and contribute to the main discussion. In fact, the authors state that “fitness tests showed antibiotics treatment negatively affected the development and reproduction in the treated generation (feeding antibiotics during the larval stage) and their untreated offspring (without feeding antibiotics).” However, the way the data are presented in Table 1 makes it very difficult to understand the changes under the antibiotic condition. My recommendation is to highlight the parameters in which the authors observed differences between the antibiotic vs. non-antibiotic groups, particularly in comparison with the controls.

Our answer: We agree and have now presented Table 1 by a figure (now see Fig. 4). We also kept the data of Table 1 in Table S1 (b) for reference. Moreover, we have now enriched the discussion on this issue, which further enhanced the interest and value of this study.

Discussion section. It is important to emphasize in the Discussion section that the antibiotic treatment was used to disrupt the gut microbiota composition. This is only briefly mentioned in the Methods section, yet the effects of the treatment are clearly shown in Figures 3 and 4. Interestingly, the control population treatment (CKa) in Figure 4 seems to be more affected than LT1a and LT2a. This result is very interesting, but is not discussed. Could it be possible that cold-exposed populations are more resilient to antibiotic treatment, or that their microbial communities have already adapted and are therefore less prone to disruption?

Our answer: We agree and have now enriched the discussion on this issue (now see the last second paragraph of Discussion), which also substantially enhances the interest and value of this study.

Figure 1a (Developmental period of larvae graph). The bars showing information for L7 appear to be significantly different, but the green bar (Control group, CK) is denoted with the same letter “a” as the blue bar. Are both truly similar? It would be helpful to clarify this and to add in the figure legends which statistical tests were applied for L1–L3 and which were applied for L4–L7.

Our answer: We agree and have now corrected this error. We also provided the statistical methods in the figure caption now.

Line 60–61. In the text “heat shock Hsp expression”: Do the authors mean heat shock proteins (HSPs)? It is important to define the abbreviation here and to use it consistently throughout the text. Is the abbreviation in line 97 referring to the same proteins?

Our answer: Agree and revised here.

Line 256. Misspelling of “Temperature.”

Our answer: Thanks and corrected.

Line 355. Misspelling of “Present study.”

Our answer: Thanks and corrected.

Reviewer 3 Report

Comments and Suggestions for Authors

Song et al. used a multigenerational cold acclimation design and 16S rDNA sequencing, they investigated the possible pattern of cold acclimation and contribution of gut bacteria in Spodoptera frugiperda. The manuscript is well-written, and the research work is solid. I recommend accepting it for publication after minor revisions.

  1. Materials and Methods (Line 148): Why is low-temperature treatment required to be conducted in a dark environment? Does the dark environment have any other impacts on the experimental group?
  2. Please try to keep the arrangement order of the pictures consistent throughout (all arranged from top to bottom OR from left to right); for the figure captions, please annotate the content of each small picture separately, and do not merge a large number of them into one.
  3. In the Discussion section, the analysis of how antibiotics disrupt the balance of its intestinal microbiota can be more in-depth. If possible, it would be better to combine this with the mentioned changes in the abundance of cold-tolerant bacteria after antibiotic treatment; meanwhile, this part can also be used to put forward some prospects for pest control.
  4. A summary paragraph of the entire article can be added at the end of the text.

Author Response

Comments and Suggestions for Authors

Song et al. used a multigenerational cold acclimation design and 16S rDNA sequencing, they investigated the possible pattern of cold acclimation and contribution of gut bacteria in Spodoptera frugiperda. The manuscript is well-written, and the research work is solid. I recommend accepting it for publication after minor revisions.

Our answer: We are very grateful for the positive comments and constructive suggestions to our MS. We have now revised the MS very carefully accordingly to all these comments and suggestions from all the three reviewers.

  1. Materials and Methods (Line 148): Why is low-temperature treatment required to be conducted in a dark environment? Does the dark environment have any other impacts on the experimental group?

Our answer: Spodoptera frugiperda is a noctuid moth. For the convenience of rearing and treatments, we set 9:00am to 8:00pm as scotophors. We thus take cold treatment during the dark stage. This information has now been added in the first two paragraphs of the methods part.

  1. Please try to keep the arrangement order of the pictures consistent throughout (all arranged from top to bottom OR from left to right); for the figure captions, please annotate the content of each small picture separately, and do not merge a large number of them into one.

Our answer: We thank these comments and have now revised the captions accordingly. We did not change the order of the subgraphs in pictures due to the consideration of both the arrangement and appearance of each figure.

  1. In the Discussion section, the analysis of how antibiotics disrupt the balance of its intestinal microbiota can be more in-depth. If possible, it would be better to combine this with the mentioned changes in the abundance of cold-tolerant bacteria after antibiotic treatment; meanwhile, this part can also be used to put forward some prospects for pest control.

Our answer: We agree and have now provided more in-depth discuss on this issue. We also discussed the potential usage of antibiotic treatment on pest control.

  1. A summary paragraph of the entire article can be added at the end of the text.

Our answer: A summary paragraph was provided now at the end of Discussion.

Round 2

Reviewer 1 Report

Comments and Suggestions for Authors

The authors have attended most of my previous comments, but some flaws still remain and need to be addressed:

Line 126: This paragraph starts stating that “Based on the above achievements on temperature stress and adaption mechanisms in insects...”. However, the last sentence of the previous paragraph is going against the hypothesis presented here. Please check the context.

Line 147: Figure 1 legend should include the details of what is being presented to be self-informative. The text that appears at the bottom of the figure must be part of the legend, but the word “Measured” after “Measurements:” should be removed.

Line 264. The authors indicate that their results indicate that cold tolerance has a “positive correlation with two factors: lower exposure temperatures and a greater number of generations”. However, is there is a common increase of two variables, the correlation is positive (cold tolerance and number of generations) but if one increases while the other decreases the correlation is negative (cold tolerance and temperature). By the way, it is not clear if “lower exposure temperatures” are “lower time of exposure” or “lower temperature”, as both parameters have been tested (11ºC during 8 or 12 h, and 11 or 9 ºC).

L311 and 319: It is not clear what the “observed features” refer to.

Line 353: I still consider that the functional prediction based on BugBase is not informative and I suggest to remove the references to it in the manuscript.  

Line 365: CKa, LT1a, LT2a are not controls. Please rephrase the statement.

Lines 383-385: I do not understand this explanation. It is not clear how reduced growth rate results in heavier and larger individuals.

Lines 469-471: The sentence needs to be rephrased.

Figures 2 and following. At the end of the figure legend it should be stated that the color code used mirrors that of figure 1, to identify the different lines analyzed.

Comments on the Quality of English Language

English usage has been improved, but needs more work. I do not intend be exhaustive, but below are a bunch of modifying suggestions to be considered. Yet, I recommend to ask for professional advice to a native English speaker.

Writing changes suggested:

Line 34: “Biodiversity studies through 16S RNA sequencing” instead of “Sequencing”

Line 36: “compared to controls” instead of “but lower abundance in controls”

Line 39: “valuable” or “useful” instead of “invaluable”

Line 39: “valuable” or “useful” instead of “invaluable”

Line 55: “shown” or “demonstrated” instead of “exhibited”

Line 56: “a few generations of cold” instead of “a few generations’ cold”

Line 71: “preventing” instead of “defending”

Line 72: “Thus, in fruit flies some Enterococcus species (e.g., E. faecalis), had detrimental impacts on reproduction” instead of “Such as some Enterococcus species (e.g., E. faecalis), had detrimental impacts on the reproduction of fruit flies”

Line 76: Remove “of fruit flies”, as it is redundant.

Line 79: “by it” instead of “by microbiota”, as microbiota is already mentioned in the same sentence.

Line 83: “of their hosts” instead of “of hosts”

Line 91: “A study on bees also revealed” instead of “A study on bees further reveals”

Line 92: “of the gut microbiota” instead of “of gut microbiota”

Line 100: “of their aphid hosts” instead of “of aphid hosts”

Line 102: “their cold tolerance” instead of “the cold tolerance of flies”

Line 104: “in these flies” instead of “in flies”

Line 144: “This study involved seven consecutive generations” or “This study was carried out over seven consecutive generations” instead of “This study conducted seven consecutive generations”

Line 159: “Measurement of development and fitness” instead of “Measured the development and fitness”

Line 159: “We used 60 individuals per sample” instead of “Each sample used 60 individuals”

Line 161: “We measured” instead of “Measure”. Same in line 163. In this subsection, there are many sentences lacking a subject (Collect, Record) as if it were a lab protocol instead of a manuscript to be published.

Line 188: “lines” instead of “above”. The same in Line 197.

Line 276: “had” instead of “showed”

Line 277: “Antibiotics had no apparent effects” instead of “Antibiotics did not show obvious effects”

Line 302: “all of them reached a plateau” instead of “the final trends of species dilution curves tend to be parallel”

Line 440: “relative abundance” instead of “order”

Author Response

Comments and Suggestions for Authors

The authors have attended most of my previous comments, but some flaws still remain and need to be addressed:

Our answer: We are very grateful for the reviewing of our MS again.

Line 126: This paragraph starts stating that “Based on the above achievements on temperature stress and adaption mechanisms in insects...”. However, the last sentence of the previous paragraph is going against the hypothesis presented here. Please check the context.

Our answer: We agree and have now deleted this sentence because it may not relevant to focus of this study.

Line 147: Figure 1 legend should include the details of what is being presented to be self-informative. The text that appears at the bottom of the figure must be part of the legend, but the word “Measured” after “Measurements:” should be removed.

Our answer: We agree and have now revised the figure and provided more information to the legend.

Line 264. The authors indicate that their results indicate that cold tolerance has a “positive correlation with two factors: lower exposure temperatures and a greater number of generations”. However, is there is a common increase of two variables, the correlation is positive (cold tolerance and number of generations) but if one increases while the other decreases the correlation is negative (cold tolerance and temperature). By the way, it is not clear if “lower exposure temperatures” are “lower time of exposure” or “lower temperature”, as both parameters have been tested (11ºC during 8 or 12 h, and 11 or 9 ºC).

Our answer: Thanks and have now revised the sentence to: “It is also noticed that the cold tolerance of the cold-exposed lines increased with lower exposure temperatures, longer individual exposure durations, and more treatment generations.”

L311 and 319: It is not clear what the “observed features” refer to.

Our answer: Agree and have now stated as: “observed features (number of ASVs)”.

Line 353: I still consider that the functional prediction based on BugBase is not informative and I suggest to remove the references to it in the manuscript.

Our answer: We agree that the BugBase prediction is not so informative. However, we still suggest to keep its references due to two considerations: 1) BugBase results further revealed remarkable differences among treatments (see the 4th paragraph of Discussion); 2) BugBase prediction showed the highest Forms_Biofilms function in LT1 cold-acclimated populations, which may be relevant to resistance (Qiao et al. 2020; Wu et al. 2021). These results and discussion may provide useful information for future research.

Line 365: CKa, LT1a, LT2a are not controls. Please rephrase the statement.

Our answer: Agree and revised.

Lines 383-385: I do not understand this explanation. It is not clear how reduced growth rate results in heavier and larger individuals.

Our answer: Thanks and more information on this issue was provided now in this paragraph.

Lines 469-471: The sentence needs to be rephrased.

Our answer: Done.

Figures 2 and following. At the end of the figure legend it should be stated that the color code used mirrors that of figure 1, to identify the different lines analyzed.

Our answer: Agree and done.

Comments on the Quality of English Language

English usage has been improved, but needs more work. I do not intend be exhaustive, but below are a bunch of modifying suggestions to be considered. Yet, I recommend to ask for professional advice to a native English speaker.

Writing changes suggested:

Line 34: “Biodiversity studies through 16S RNA sequencing” instead of “Sequencing”

Line 36: “compared to controls” instead of “but lower abundance in controls”

Line 39: “valuable” or “useful” instead of “invaluable”

Line 39: “valuable” or “useful” instead of “invaluable”

Line 55: “shown” or “demonstrated” instead of “exhibited”

Line 56: “a few generations of cold” instead of “a few generations’ cold”

Line 71: “preventing” instead of “defending”

Line 72: “Thus, in fruit flies some Enterococcus species (e.g., E. faecalis), had detrimental impacts on reproduction” instead of “Such as some Enterococcus species (e.g., E. faecalis), had detrimental impacts on the reproduction of fruit flies”

Line 76: Remove “of fruit flies”, as it is redundant.

Line 79: “by it” instead of “by microbiota”, as microbiota is already mentioned in the same sentence.

Line 83: “of their hosts” instead of “of hosts”

Line 91: “A study on bees also revealed” instead of “A study on bees further reveals”

Line 92: “of the gut microbiota” instead of “of gut microbiota”

Line 100: “of their aphid hosts” instead of “of aphid hosts”

Line 102: “their cold tolerance” instead of “the cold tolerance of flies”

Line 104: “in these flies” instead of “in flies”

Line 144: “This study involved seven consecutive generations” or “This study was carried out over seven consecutive generations” instead of “This study conducted seven consecutive generations”

Line 159: “Measurement of development and fitness” instead of “Measured the development and fitness”

Line 159: “We used 60 individuals per sample” instead of “Each sample used 60 individuals”

Line 161: “We measured” instead of “Measure”. Same in line 163. In this subsection, there are many sentences lacking a subject (Collect, Record) as if it were a lab protocol instead of a manuscript to be published.

Line 188: “lines” instead of “above”. The same in Line 197.

Line 276: “had” instead of “showed”

Line 277: “Antibiotics had no apparent effects” instead of “Antibiotics did not show obvious effects”

Line 302: “all of them reached a plateau” instead of “the final trends of species dilution curves tend to be parallel”

Line 440: “relative abundance” instead of “order”

Our answer: We are deeply grateful for the detailed and constructive revision suggestions. In response, we have thoroughly reviewed the manuscript and made revisions to all the indicated sections, along with other parts where improvements were deemed necessary. The language of the manuscript has also been polished for clarity and accuracy. It was proofread and refined by Prof. Hui Ye from Yunnan University, an expert in Entomology with international academic experience. Additionally, the grammar of the manuscript was checked using Grammarly (https://app.grammarly.com) to ensure further linguistic quality.

References:

Qiao, J., M. Zhu, Z. Lu, F. Lv, H. Zhao, and X. Bie. 2020. The antibiotics resistance mechanism and pathogenicity of cold stressed Staphylococcus aureus. LWT 126:109274.

Wu, S., Y. Yang, T. Wang, J. Sun, Y. Zhang, J. Ji, and X. Sun. 2021. Effects of acid, alkaline, cold, and heat environmental stresses on the antibiotic resistance of the Salmonella enterica serovar Typhimurium. Food Research International 144:110359.